# The Multicultural Church of "Le Jour du Seigneur"

**Pierre Hegy** 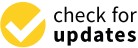

Sociology Deparment, Adelphi University, Garden City, NY 11530-0701, USA; pierre.hegy@gmail.com

**Abstract:** Multicultural worship is defined here as a form of worship that is attractive to both non-religious outsiders and religious insiders. It is most appropriate in our times of religious decline. This paper presents a Catholic television program which involves collaboration with Protestants, the secular state television, secular writers, and university professors. This Sunday service consists of two parts: a discussion called "le magazine" and the mass taking place every week in a different parish. During the pandemic, when there were strict restrictions from March 2020 to September 2020, the program aired innovative worship services, centered on music and images, broadcast from a small Paris studio. When in September 2020 the pandemic was thought to be over and the major restrictions were lifted, the program became theologically and pastorally more multicultural than ever before. The conclusion offers other examples of multicultural worship adapted to our times of religious decline.

**Keywords:** Catholic cyber church; lay-centered program; multicultural worship

This paper describes the multicultural service of the Catholic television program, *Le Jour du Seigneur*, on the French state television. It is descriptive for the benefit of liturgists. It endeavors to be theologically neutral and wants to avoid theological arguments. My thesis that in our times of religious decline this program offers alternatives to the traditional monocultural services. Its organizational structure is unique: it is run by lay specialists working in harmony for members of a religious order. Each Sunday the program begins with a general discussion about a pressing social issue. The Sunday mass takes place in a different church every week which allows for variation. During the pandemic it showed exceptional creativity in worship and music. After the pandemic it emphasized the contribution of lay volunteers in the local churches. It is highly successful: each week it attracts about half a million viewers through television and the internet. Let me begin with a definition of multiculturalism.

## 1. Introduction: Monoculturalism vs. Multiculturalism in Worship

Mainline Protestantism and Catholicism have a long tradition of monocultural worship because their liturgies are strictly regulated by official church documents, such as the Book of Common Prayer or the Roman Missal. Before Vatican II, the Latin Mass was almost the same all over the Catholic world[1]. The Vatican II liturgical reforms, which sought to overcome this monoculturalism, experienced strong resistance—a resistance that has continued to this day. Archbishop Marcel Lefebvre opposed most Vatican II innovations. He created the Society of Saint Pius X from which emerged the Society of Saint Pius V, the Priestly Fraternity of Saint Peter, and numerous other resistance groups.

Today we witness a return religious sectarianism. In the United States, the religious issues of abortion and same-sex marriage are divisive political issues. On the international scene, Islamic terrorism propagates violence in order to establish a dictatorial form of Islam. In the Middle East, where Christians and Muslims have cohabited peacefully for centuries, the Shiites and the Sunnis kill one another. Of course, these conflicts also involve economic and ethnic tensions. In the Far East, one Buddhist nation persecutes the Muslims, elsewhere the Hindus persecute the Muslims, and elsewhere the Muslims persecute Christians. Within

Catholicism, traditionalists favor a return to pre-Vatican II forms of worship and reject, often in violent terms, what they see as unacceptable individual liberalism.

In the context of religious decline in the West, which is characterized by increased religious indifference of the masses and increased sectarianism of conservatives, I see multicultural worship as a form of worship that is attractive to outsiders while also satisfying insiders. It is a form of adaptation in which the outsiders are not coopted by the insiders, and the beliefs of the insiders are not watered down to the outsiders' lowest common denominator. Many English-speaking churches have services in Spanish or another foreign language, but this multiculturalism only adjusts to internal differences and does not extend beyond the borders of the church or denomination. What is needed is a new form of evangelism that is respectful of the outsiders' particularities while remaining loyal to the insiders' religious identity.

This paper presents an example rather than a theory of this form of multiculturalism. It is purely descriptive in order to avoid theological and theoretical disputes. It is geared to liturgists whose goal is to implement vital forms of worship, rather than theologians for their doctrinal evaluation. The example presented here is highly successful, which suggests that there is something valuable in it. Some American megachurches are good examples of worship attractive to outsiders, such as the seeker-friendly Community Church of Willow Creek and the purpose-driven Saddleback Church. They uphold different (and maybe questionable) theologies, but they are successful for reasons other than their theologies.

The example of multicultural worship described here is the Sunday worship program called *Le Jour du Seigneur*. It has the following characteristics. (1) Many of its programs are Protestant–Catholic co-productions on topics of common interest. (2) It is independent from yet maintains a friendly relationship with the French Catholic bishops and thus its orthodoxy cannot be questioned. (3) For the last 70 years it has worked in collaboration with the secular French State and remains respectful of French secularity but untainted by secularism. (4) The first part of the program discusses social, political, economic, and cultural issues of interest to non-believers, yet from a Christian perspective. (5) It offers the Sunday mass for believers, but each week in a different parish which stimulates interest. (6) During the pandemic, while most churches streamed lifeless liturgies in empty churches without music, this program created worship of exceptional artistic quality. (7) It is highly successful, attracting half a million viewers each week through television and the internet.

*Le Jour du Seigneur* (LJS) airs once a week on Sundays from 10:30 a.m. to 12:00 p.m. by the French State television station, France-2. The mass is broadcast live Sunday morning for television viewers, and in recorded form a few hours later for internet viewers. Apart from worship, it involves multiculturalism in ethnicity, art, music, and general culture. It is ethnically and culturally diverse since the program can take place in any French-speaking country in Africa, the Middle East, and the New World. There is a variety of social media platforms where viewers can stream the event; half the program takes place in the Paris television studio and the other half each week in a different parish anywhere in the world. Musically, it is contemporary; drums and other modern instruments are more common than the traditional organ. Intellectually, it is diverse—the topics discussed in the first half of the program range from spirituality, psychology, social problems to politics, and more. Organizationally, the program is operated by a lay organization independent of the Catholic hierarchy but assisted by members of the Dominican order. Although it is aired once a week, this program attracts over half-a-million viewers each week. How can this program be so popular in a highly secularized and mainly agnostic country? The main reason for its success seems to be its multicultural dimension.

## 2. Past Practices and Theories of Multicultural Worship

Throughout the Middle Ages, devotions were mainly local, consisting of processions, pilgrimages, vigils, and the celebration of a saint's day. Some were individual or family practices, but many were organized by local lay confraternities that were independent of the parish clergy. Some of these confraternities still exist in Latin America (Rojas Lima

1988). It was a situation of geographical pluralism due to the lack of communication and centralization. After Trent, the papacy encouraged an ever-greater number of devotions. The first major innovation was the feast of Our Lady of the Rosary by Pope Pius V after the naval victory of Lepanto in 1571. In the 19th century, the practices related to the Eucharist became more prominent, namely the Forty Hours before the Blessed Sacrament, nocturnal adoration, devotion to the Sacred Heart, and the novena of First Friday masses. The popes granted indulgences to encourage these devotions. By the middle of the 19th century, most popular devotions had become clergy-centered (Francis 2014, p. 132). In the United States, the period from the end of World War I to the mid-1950s was the heyday of American devotionalism (Chinnici 2004, p. 52). Traditional devotions (to St. Anthony, the miraculous medal, and the scapulars (Traves 1986, p. 38)) and new ones (St. Jude, the novenas to Our Lady of Perpetual Help, and to Our Sorrowful Mother) attracted ever more fervent prayers. Two new religious movements—the enthronement of the Sacred Heart in homes and the common recitation of the rosary—made devotions part of family life. Launched in about 1943, the movement had achieved 500,000 enthronements by 1946 (Chinnici 2004, p. 61). At about the same time, Fr. Patrick Peyton propagated the rosary crusade on radio. It achieved exceptional success, with mass rallies in the U.S. and Canada. While devotions were now centralized and uniform, there was the multiculturalism of religious devotions on the one hand, and the official liturgy on the other.

While devotions were growing in the U.S., the liturgical movement in Europe called for the primacy of the liturgy, away from the centrality of devotions. One main promoter was Dom Guéranger, the abbot of the Benedictine Abbaye of Solesmes, whose primary function has always been the celebration of the liturgy. This spirit prevailed at Vatican II and was enshrined in its constitution *Sacrosanctum Concilium* (1963), calling for "the restoration and promotion of the sacred liturgy [and] full and active participation by all the people" (#14). It required that devotions "be so drawn up that they harmonize with the liturgical seasons, accord with the sacred liturgy, [be] in some fashion derived from it, and lead the people to it, since, in fact, the liturgy by its very nature far surpasses any of them" (#13). Private devotions were devalued since the liturgy "far surpasses any of them". Within a decade or two, for a variety of reasons, devotions faded away in Catholic life. Since the 19th century, the clergy had taken control over most popular devotions, but now it progressively withdrew its support. This situation created a "piety void" in the life of many American Catholics. There increasingly came to be a gulf between traditional devotions that had inspired the faith of many generations, and the new liturgy which was "not yet sufficiently meaningful or satisfying to fill the void left by pious devotions". (Chinnici 2004, p. 82). The current situation today is a strange form of religious multiculturalism. One the one side, we find the majority of Catholic and Protestant believers who do not attend church worship or who campaign for a return to traditional church services, and on the other we have the public worship in declining churches which hold on to traditional monocultural liturgies. What is needed is to overcome past theological divisions and offer new forms of worship attractive to both insiders and outsiders.

At the theoretical level, there is the fundamental question of who owns, or who is in charge of, worship. Is it the body of believers, as suggested by the Greek meaning of *leitourgia*, or the church authorities? The Protestant denominations solved the problem by adopting either a denominational or an episcopal polity, or a mixture of the two. In the Catholic Church, there was a long tradition of local pluralism during the Middle Ages, which was followed by the monocultural worship initiated by the Council of Trent.

I will present an alternative to this monocultural type worship by presenting, first, the general rule of *lex orandi lex credendendi* (the law of prayer is the law of faith) and, second, two interpretations of it by Aidan Kavanagh and Kevin Irwin. Finally, I will present pragmatic solutions devised by innovative pastors.

The maxim of Prosper of Aquitaine (390–455), *Lex orandi lex credendi*, has been given two opposite interpretations. I will leave aside the historical setting (see Novak 2014) and avoid taking sides in this theological debate. From a monocultural perspective, the law of

prayer is the official liturgy and the law of belief is that of the official church. This was the position of Pope Pius XII, who wrote, "The epigram, 'Lex orandi, lex credendi'—the law for prayer is the law for faith … this is not what the Church teaches and enjoins". The law of belief is that of the Catholic tradition as taught by the Magisterium. Hence Prosper's maxim is reversed. According to the pope, the "perfectly correct" position is that "*Lex credendi legem statuat supplicandi*—let the rule of belief determine the rule of prayer". Pius XII made this point in his important encyclical on the liturgy, *Mediator Dei* (#46–49). This position is in fact implemented universally because the Catholic liturgy is that of the official missal. A rigid understanding of the pope's position leaves little room for multiculturalism. However, there are theologians who disagree.

In his treatise *On Liturgical Theology*, Aidan Kavanagh took the opposite position. He posited that prayer leads to belief, or, in his terms, "worship conceived broadly is what gives rise to theological reflection" (Kavanagh 1984, p. 3). This view is based on two assumptions: one, that liturgy produces "deep change in the lives of those who participate in [it]", and two, that this change leads to theological insights called *theologia prima*, from which academic theology or *theologia secunda* is derived. For Kavanagh, these assumptions are observable realities that he has witnessed "in a fairly regular way … all over the world" (Kavanagh 1984, pp. 73–76, 93). This view is both attractive and challenging. It is based more on personal conviction than theological arguments, being more mystical than academic.

A middle of the road position is presented by Kevin Irwin in *Context and Text: A Method for Liturgical Theology* (Irwin 2018). His method is one of contextual analysis. In his view, the liturgy is an event that varies geographically and historically rather than a text. Throughout his book, Irwin struggles with the various interpretations of Prosper's maxim. Instead of siding with one interpretation or the other, he added a new dimension, that of moral life, the *lex vivendi.* Now, *lex orandi, lex credendi, lex vivendi* becomes interactive rather than linear—i.e., prayer, beliefs, and moral life interacting with one another. This 650-page proposal is well balanced, but it will have little effect outside the circle of liturgical theologians. Hence, we have no universally accepted theoretical solution.

A third approach to the opposition between monocultural and pluricultural worship is pastoral and pragmatic. This position is not defended by intellectuals or organizations but is applied when the pastor sides for one position or the other or both. This is usually done without theological, canonical, or conciliar arguments. A basic fact of church life is that bishops and pastors can enjoy a fair amount of freedom when they want to or are innovative. Thus, Catholic bishops can abolish or change the structure of parishes without permission from Rome. Any parish pastor knows that the liturgy of the Eucharist is strictly regulated while that the liturgy of the Word is not, and non-liturgical services are wide open to innovations. Thus, a conservative innovation at a Catholic mass could begin with the recitation of the rosary and end with more prayers and another sermon. A progressive innovation could begin with a general discussion of the scriptural readings of the day, and end with a business meeting about church ministries. This pastoral conception of worship is what we find at *Le Jour du Seigneur*.

## 3. Practices of Multicultural Worship: The Structure and the Programs of the CFRT

*Le Jour du Seigneur* was created in 1949 by the Dominican Raymond Picard as a 90-min program on Sunday morning. It is aired on the state television France-2, in a country where half of the population does not believe in God, at least according to a recent survey by IFOP (*Le Monde* 2021) and Catholic weekly church attendance is in the single digits. The French state is secular or even secularist; it will support only religious programs that are non-sectarian and have some cultural value. LJS is ecumenical; it includes Protestant programs, usually in the form of Protestant–Catholic co-productions. Moreover, once or twice a year it holds an inter-religious conference that includes Christians, Muslims, Jews, Hindus, Buddhists, and atheists. LJS is run by a lay committee, thus avoiding the criticism

of clericalism. There is good collaboration with the state television which in fact covers half the production costs.

The great innovation is the inclusion of discussions into the Sunday worship. The 90-min program is divided into two parts: the first is called *le magazine* and the second is the mass. As in secular magazines, any topic can be brought up. Thus far, we have noticed several multicultural dimensions. It is a religious program for a public that is mainly non-religious. It is produced in collaboration with the secular state. The first half of the program, the magazine, is unregulated by Catholic canon law, hence open to innovation. The second half, the mass, is actually integrated into the magazine as we will see; hence there is no secular vs. sacred dichotomy, but a mutual integration of the two into a pluricultural perspective.

At its highest level, this program consists of two organizations: the CFRT (*Comité français de radio-télévision*), which produces religious videos, and LJS, which is the 90-min Sunday service using some of these videos. I will now present the organization of the CFRT and three of its productions, and in the next section the content of the Sunday program.

The CFRT has nine members, which includes three Dominicans. It is the ultimate authority of the organization. It can only function productively if there is collaboration between lay and clerical views. The stated mission of the CFRT is "to announce on television the message of the Gospel and to answer, in its own way, the quest of meaning of our contemporaries".[2] It presents itself as Christian rather than Catholic. The committee is independent of the Catholic hierarchy since it is lay and not clerical. Its mission is to share the values of the gospel, not to teach Catholic doctrine. Its programs must be creative but avoid controversial topics. It can offer a critique of social institutions, but without antagonizing the state, politicians, or the Catholic hierarchy. In this description we have noticed a few more multicultural practices: cooperation between clergy and laity, Christian rather than Catholic identity for greater intercultural dialogue with the secular public, non-polemical criticisms of the state and the church in implicit or explicit collaboration.

The CFRT is a production company of about 60 employees that creates audio and video material to be used in Catholic and Protestant schools and churches, or to be aired on French-speaking televisions stations in Europe and Africa. These productions are available at Vodeus.com. They are divided into *folders* regarding a specific topic (e.g., cathedrals, the Camino to Compostela, Christian–Jewish and Christian–Muslim dialogues, the effects of slavery, young people and faith—each of these may consist of several videos), *documentaries* (e.g., the Catholic Church in China and in various African nations, stained-glass windows, the American church in Paris, Kairos, Catholic vs. Protestant preaching, Taizé, etc.), and also *series*, i.e., topics developed in many videos (on icons, art and faith, St. Paul's missions, the bible in comic strips, Mary in the bible, the saints, the sacraments). I will present two short videos (Unexpected Words and The Pillars of Notre Dame) and a long documentary (Easter in Art).

The title *Parole inattendue* (Unexpected Words) is puzzling, maybe intentionally. It consists of interviews of personalities of the media during a two- to three-minute taxi ride in Paris[3]. The interviewees were asked to select a verse from the Sunday readings and offer comments, and explain their views about God and faith. Over 60 personalities have accepted so far. One would expect to find mainly Catholic intellectuals, but this is not the case. Among them we find quite a few writers but also several rap artists, a film maker, a theater producer, a professor of philosophy, a physicist, several singers, a caricaturist, a female bicycle runner, a professional football coach, a high-level state administrator, a psychoanalyst, a psychiatrist, a comedian, and a few more. Nearly all avoided commenting on the verse they had selected. A few, however, were moving. A non-practicing Jew selected, "My God, why have you abandoned me?" It represents the history of his life: his father, a member of the French resistance died in Auschwitz. Orphan at age 11, he turned to God at night, "God, God, are you here? Why have you taken away my parents? Why?" Equally moving were the testimonies of a two or three practicing Christians. Most of the interviewees were non-practicing, agnostic, a few were pantheists or atheists, but

none was anti-religious. All expressed having some faith, most of them in humanity. None mentioned Jesus Christ or salvation.

The puzzle of this series is why they were shown *after* mass like a continuation of the mass and its preaching. Maybe the producers simply wanted to show that they are listening to the people, whatever they say. This series can also be understood in the light of an important statement presented below, that the Holy Spirit is at work in all people. Can this be true of non-believers? The logic of the program leads one to believe so.

*Les Piliers de Notre Dame* (the Pillars of Notre Dame[4]) is another series presented at the end of the mass, besides *Parole inattendue*. There is national interest in the rebuilding of the Notre Dame cathedral after the fire of 2019. In each segment, we learn from an artisan or building expert about their work. All are proud of their achievements, and many see their work as a spiritual mission. Church renovation is a small national industry, as an estimated 45,000 French churches are maintained, partially or totally, by public funds[5]. The purpose of the series seems to be that all people through their professional work can contribute to the kingdom of God, because the Holy Spirit is at work in all people, even in their secular work. Here is one more multicultural dimension: the work of the Holy Spirit and the kingdom of God extend into secular work, even among non-believers.

On Easter Sunday of 2020, the documentary *L'Art de la Pâque* (Easter in Art[6]) was shown. The film was produced by the CFRT in collaboration with the state television. It included a Byzantine fresco, the paintings of Giotto, and those of Gruenewald; the baroque art inspired by Tridentine theology was not mentioned. The commentators of these works included an orthodox theologian, a historian from the university Paris-1, a female historian, a female pastor and theologian, and a Jesuit art historian. Here is what they had to say about the Byzantine fresco in the form of a quilt of quotations. "This Byzantine fresco shows Christ pulling humankind out of its graves. Christ descended into hell to grab Adam and Eve and take them with him because hell could not contain them anymore. Here is why he came: to take humankind with him. This going down into hell is the main theme of orthodox iconography; there can be no other. It shows the movement of Jesus going into the world of death to grab our lives to lead them to plenitude. We are at the center of a story of emancipation, of liberation from all forms of death, physical and spiritual. At the bottom of the fresco is Satan, Evil, the Enemy who now has not more power of humankind. Christ went to find what was dead, the old self which is to be reborn. What is suggested here is a new birth, a new baptism [in the spirit]". This is the theology of divinization or *theosis*, the heart of orthodoxy; it is the equivalent of the spirituality of sanctification in the West. The comments about the paintings of Giotto and Mathias Gruenewald are equally inspiring. What is significant here is that the baroque art inspired by the reform of Trent is not mentioned. We go back to pre-Reformation times to rediscover the universal messages of Giotto, Gruenewald, and Orthodoxy.

In summary, we noticed a few more intercultural practices. Most generally, we observed that the videos produced by the CFRT are multicultural, addressing a variety of publics: schools and churches, Catholics and Protestants, national and foreign television networks. The Easter video was created through the collaboration of academicians and pastors of various churches, Orthodox and Catholic. *Parole inattendue* invited personalities of the media to comment on the Sunday readings; they gladly accepted but ignored the readings, presenting their secular religion instead. The Pillars of Notre Dame glorified professional work, religious and secular. There seems to be no limit to multiculturalism when worship includes pre-evangelism, which is the purpose of the magazine. We must now turn to the content of LJS.

## 4. The Weekly Multicultural Worship

According to a French copyright law, the programs of the state television cannot be recorded for later public transmission. Hence, the past Sunday services of LJS are not available. One can, however, record them privately for personal use. I recorded 13 Sunday services between 2016 and 2017, five in March and April during the Covid shutdown, and

11 starting in September 2020, when the pandemic restrictions were slightly lifted. My presentation is limited to these recordings, plus personal notes about other Sundays.

Before the pandemic, the magazine and the liturgy were usually well integrated around a theme, but this was not possible during the pandemic. I will first present the content of the magazine, and next the Sunday worship.

*4.1. The 2016–2017 Sunday Services*

To call the first part of LJS a discussion would be misleading; it would suggest a debate about an intellectual topic, but this is seldom the case. Worship is mainly a community celebration; any community event can be the theme of a given Sunday. One moving example was a mass celebrated in a small chapel—maximum capacity of 20 people—which had been built by homeless people 50 years ago. They were invited to come back and offer their testimonies—that was not a discussion. I will give three examples of pre-pandemic services, from the most local to the most universal.

On 15 October 2016, the Sunday service took place in the Republic of Mauritius, a tiny island in the Indian Ocean, about 1200 miles east of Africa. The religion of the majority of Mauritians is Hinduism, the official language is English, the common language is Mauritian Creole, and only 2% of the population speaks French. The reason for the report was the not yet announced nomination of its local bishop, Mgr. Maurice Piat, as cardinal just days before the visit to Mauritius, as if LJS had received an advance notice through a back channel. In the long introduction about the history of the island, we learned about slavery introduced by the French governor François de la Bourdonnais and the plantation economy of the island. When slavery was abolished, the slave work of Africans was replaced by the exploitation of immigrants from India. During the first half of the documentary, it seemed that its purpose was to discuss French slavery; it is only in the second half that it became clear that its message was the work of the local bishop and the local church to alleviate the local consequences of slavery. In his first sermon as a cardinal elect, Mgr Piat did not preach about the pope's doctrine of social justice but about what his local church had been doing and what remained to be done[7]. This sermon was a local as local can be.

The 2016 anniversary mass for the 130 victims of the Bataclan massacre was not even about a local church but about the Parisian neighborhood where the massacre took place in 2015. The interview with the pastor of St. Ambroise where the Bataclan theater was located took place in the café whose owner was killed in the massacre. The interview recounted, in words and images, the aftershock among neighbors, and their spiritual struggle to cope with the disaster. The priest mentioned the Holy Spirit at work to bring unity in love. It was a message of hope to overcome anger and the sadness of death. The mass was celebrated in great solemnity with the participation of 8 priests and the presence of an imam.

There was also a universal dimension. On that day, the magazine presented a 26-min documentary entitled *Un Chemin vers la Lumière* ("A Road towards Light"), which showed the testimony of Fouad Hassaun in his struggle toward the light of forgiveness[8]. On 21 January 1986, a bomb exploded in a Christian neighborhood of Beirut, killing 30 people. Among the 250 injured was Fouad, a 17-year-old medical student. After an 11-h operation, he was taken to a hospital in Switzerland where he was declared permanently blind. Alone, he started his long struggle with God to find hope. "No, Lord, I cannot accept what is unjust; I refuse". He accepted, but it took him over 10 years to find the peace of forgiveness and overcome the desire for revenge. Yes, forgiveness is possible. Through it, Fouad found inner peace, happiness, and marriage. He returned to Beirut to revisit the scene of the massacre and testify to his peace of forgiveness. He founded an association of Muslims and Christians for mutual understanding. This long documentary is an example of what the magazine stands for: testimonies, not intellectual discussions.

An example of universal concerns is the program of 15 January 2017, which discussed the general theme of how to give a human face to the migrants[9]. At the beginning of the magazine, Jesuit Fr. Thierry Lamboley gave a theological explanation. The Bible is clear: "Do not exploit or oppress the immigrant because you have been immigrants in Egypt".

(Exodus, 20:20). A video showed the refugee camp of Aleppo in Syria where volunteers treated the displaced persons with respect and dedication. We next went to the parish of St. Paul in Hem in the North of France; it has no web page, no email, and no resident priest but an exceptional ministry to the immigrants. About 40 parishioners were involved, providing shelter and work to immigrants. Next, a documentary described the muddy and sordid refugee camp of Calais were Brigitte and Olivier are called Mom and Dad. They regularly visit the camp with food and supplies. They also take migrants to their home where they can take a shower, get their clothes washed, and enjoy a home-cooked meal. These refugees attempt to reach England which closed its border to them. We next follow Brigitte and Olivier in their tour to England to visit those of their adopted children—hundreds of them, we were told—who managed to enter the UK illegally. Significantly but not mentioned explicitly: Brigitte and Olivier do not belong to any church. At St. Paul church, on the day of the migrants and refugees, the mass was celebrated with banners and festive music; the whole assembly had even memorized the Our Father in Aramaic, the language of Jesus, which was recited by all parishioners with great conviction. On that day, the magazine and the liturgy were perfectly integrated. Needless to say, in these three examples we have many more dimensions of multiculturalism: the history of slavery and its consequences in Mauritius, testimonies about healing in the anniversary mass of the Bataclan massacre, and documentaries about helping migrants.

### 4.2. The Pandemic Liturgies

Because of the pandemic curfew imposed in March 2020, the planned Sunday mass in a parish had to be canceled. What to do? The Dominicans could improvise a Sunday mass because their convent happened to be located next door to the studio of LJS. Three Dominicans spent days and nights searching and rehearsing, and the following Sunday their mass was ready. From the Parisian studio turned chapel, a new type of liturgy was aired for several weeks.

At the beginning of the Mass on March 29, Fr. Thierry Hubert came forward to the camera for an important announcement[10]. "Welcome! Our studio is small, but we will proceed as if *you* invited us to come into *your* home. We will celebrate Mass in your midst, *for you and with you*. We are all getting tired and anxious [because of the shut-down]: we now seek words and songs that nourish us, gestures that give strength and courage, and moments of communion that give comfort". A similar statement was made the following weeks. No explanation was given, but to celebrate "with you and for you" stands in contrast to the Tridentine mass "for you but without you", which requires no attendance. "With you and for you" was not theoretical or a theological agenda, but the promise of seeking words and songs that nourish, and moments of communion for comfort and unity.

In these studio masses, most of the time was taken by polyphonic singing by the three Dominicans. The quality of their singing was very high. The singing was filmed in closeup shots, which increased its impact within the confined space of the small studio. There was no musical instrument for support or background; the purity of the voices gave them a heavenly quality. Most of the pieces were original or unusual to common ears. They seemed of Byzantine inspiration (one could see a small icon of Christ on a side table). Before the Preface, the friars sang the Byzantine *Trisagion*, "Holy God, Holy Mighty, Holy Immortal, have mercy on us", and after the Preface, the *Sanctus* in French: "Holy, Holy, Holy is the God of the universe". At communion they sang verses from Psalm 33, "I will praise the Lord at all times", with the refrain, "Wisdom has set the table. She invites people to the feast. Come to the banquet of the son of man. Eat and drink the Pascua of God".

At the end of the Mass on May 3, the camera turned the viewers' gaze toward an icon of Mary during the singing of a joyful hymn, as it is the custom every Sunday at the Notre Dame cathedral in Paris. The last images of the day were the Byzantine icons of the teaching Christ, and of Mother and Child. These images of normalcy suggested that it was time to give voice again to people in the parishes, which happened as soon as the

restrictions were lessened a few weeks later. An important multicultural dimension was the use of Byzantine music and icons.

*4.3. Post-Pandemic Innovations*

Before the pandemic, the presenter, David Milliat, had often casually said, "Let us meet those women and men who have prepared the liturgy for us". This introduction took a new meaning in the post-pandemic celebrations. In September 2020, LJS went back to parishes for the Sunday celebrations. For the first four weeks, we were taken to small parishes whose characteristic, not mentioned on screen, was that they had no resident priest. Sainte Catherine in a small town in the South of France that shared a priest with six other churches. Saint Martin in the North was one of 23 villages with three priests. Saint Rémi in the East was a village of 700 people, part of a parish of 11 villages. These churches were lay centered by necessity. The parishioners were the main actors of the Sunday services, and it is their example which conveyed the message of the day. Now it makes sense to say, "Let us meet those women and men who have prepared the liturgy for us". In the first of these churches we met Analie, a 17-year-old convert who was enthusiastic about sharing God's love, as well as deacon George and the nurse Françoise who catered to various needs of the church. The priest during mass showed exceptional musical talents, but he was not introduced. In this and other churches, the singing of the choir and the assembly was enthusiastic. The simple words "let us meet those who prepared the mass" had the effect of creating a continuity rather than a dichotomy between the secular magazine and the sacred Eucharist. This continuity was reinforced by the continued presence and even the active participation of the commentator, David Milliat, during mass. His intervention on 13 June 2021 was of special significance.

It is customary at LJS for David Milliat to make comments during the liturgy. Usually, he first gives information about the mass. After the homily, he often interrupts to give the internet address where one can find the text of the homily. At communion, he usually offers a reflection. In any Catholic parish, it would be shocking if a lay person would grab a microphone and offer a reflection. On 13 June 2021, at communion in the basilica of Marseille, David Milliat reflected on the exceptional situation of Marseille as a harbor open to both Eastern Christianity and Islam in North Africa. "This situation holds a message, because the theology of the Mediterranean is one of encounter, of a relationship of friendship, inspired by the conviction that the Holy Spirit is at work in all people of whatever religious convictions". This exceptional statement must have been discussed previously and planned by the whole team of LJS. It stated that the Holy Spirit is at work not only in spirit-filled believers, but in all men and women of good will in their daily professional work, as presented in the magazine. The belief that the Holy Spirit is at work in all people is extended to Orthodox Christianity and Islam. What is remarkable is that this belief was stated by a lay person as a "conviction" rather than a debatable theological opinion. If it is a conviction of faith, who would oppose it?

## 5. Conclusion: Factors of Multicultural Worship

This paper was an implicit response to those who say, "It cannot be done. We have rules to follow". The example of LJS proves the opposite. Here are four more examples that prove how innovation was made possible.

Let us begin with what made LJS possible. The Dominican Raymond Picard was ahead of his time in his realization that the increasingly secular French society needed an ecumenical and multicultural Sunday program rather than a national Catholic television network, which came into existence in 1999. In 1949, he convinced the state television that such a program was of great cultural value, and its eventual success proved him right. Knowing that a mass only takes about 30 min, a 90-min time slot would give him 60 min for unregulated innovation. After the initial success, Picard negotiated with the state a contract for a weekly program. To shelter it from criticisms from the church and the state, he created a civil organization: the CFRT, which was composed of lay and

religious members independent from the Catholic hierarchy. In short, the major factor in favor of multicultural worship at LJS was the collaboration between the state, the church, the universities, ecumenical organizations, digital technology, interviews, documentaries, quality music, Orthodox, and Western traditions, all of which conflated amid changing times, flexibility, and adaptation.

A second example is that of the bishop of Poitiers, France, who created a new parish structure, one in which a lay team, not a priest, has the main responsibility of the parish. This team consists of five volunteers responsible for worship, teaching children, charity work, church finances, and public relations; they work in cooperation with a non-resident priest. This team is installed by the bishop in a public ceremony. All volunteers are nominated for three years, renewable only once. This innovation seems extreme: lay volunteers without experience nor seminary training must lead prayer, preaching, and teaching. This experiment stated in 1995 has been adopted in over 300 local communities. In this case, the success of the proposal was due to collaboration and diplomacy. The proposal was the creation of a local synod (1988–1993), which the new bishop, who arrived in 1994, Mgr. Rouet, was asked to implement. This required exceptional diplomacy—the bishop had to visit villages to personally seek lay volunteers for responsibilities as different as prayer and finances, teaching, and charity work, in the midst of suspicions on the part of the clergy and the populace. Here again, cooperation was key (Rouet 2005, 2008).

Another example of multicultural worship is that of the St. Sabina parish in Chicago. The pastor, Fr. Pfleger, galvanized the parish in progressively longer Sunday services that extended from one to two, even three, hours. He hired a prominent director of gospel music and adopted the praise and worship style of prayer, which calls for loud and emotional responses. Several times during the year there is an altar call as in Evangelical churches. On many Sundays, especially at Christmas, Easter, and Pentecost there is extensive choreography. In the process, many liturgical rules are ignored. Here, we have another principle that makes multicultural worship possible: the pastoral non-observance of rules judged outdated. It is not dissent; rather, it is tactical and pastoral non-compliance. It brought Fr. Pfleger into conflict with the archbishop, but ultimately he prevailed. (McClory 2010; Hegy 2017, chp. 11).

My next example takes us to the San Miguel church in Guatemala, where the success of multicultural worship was facilitated by the good relations between the pastor and the church authorities, a relationship aided by the pastor's knowledge of church rules. In 2015, this parish had about 150 small groups meeting weekly for reflections and prayer. One of their outstanding practices was a holy hour with the Blessed sacrament in the living rooms of the 150 groups. How canonical is this practice? According to church law, parish Eucharistic ministers can distribute communion at mass and bring communion to the sick. Most parishes have 10 to 15 such ministers. However, at San Miguel, the Eucharistic ministers must also visit the sick. This parish, therefore, has over 150 such ministers who can also bring the Eucharist to the 150 communities. This pastor always informs the chancery of his innovations and usually post factum. Good relations easily overcome obstacles (Hegy 2017, chp. 10).

My last example takes place in the Democratic Republic of the Congo. In 1969, the Catholic bishops of the Congo petitioned Rome for a Zairean liturgy. The soul of the request was Joseph Malula, the archbishop of Kinshasa. He also had a multicultural vision of the church. He reorganized the parishes by giving the laity a key role, even more radical than in the diocese of Poitiers. As cardinal, Malula pursued negotiations with Rome from 1974 to 1988 when, finally, a special usage of the Roman rite was granted. This Zairean liturgy takes two-to-three hours. It is highly participatory; the entrance is a choreographed procession that may take 10 min and the offering of gifts 25 min. There is constant clapping and dancing in place. The rhythmic music and the constant beat of the African drum are enthralling. The assembly is enthusiastic. At the end of this long Sunday service, many people still stay for more prayer. In this case, the success of the multicultural Zairean rite

was due to the diplomacy and perseverance of Malula through his personal friendship with Pope John Paul II. ([Hegy 2019](), chp. 9).

As a general conclusion, it seems that in a pluralistic society with declining religious interest, religion requires diversity of offerings, as in *Le Jour du Seigneur*. Besides, multiculturalism and innovations have long been the norm in education and business and are an intrinsic part of a pluralistic society. The success of LJS shows that it is on the right track.

**Funding:** This research involved no external funding; all expenses were covered by the researcher.

**Data Availability Statement:** On data availability. The material available on the internet is referenced in the footnotes given below.

**Conflicts of Interest:** The author declares no conflict of interest.

## Notes

1.   The Constitution Quo Primum of Pius V (1570) recognized the right "to celebrate Mass differently" when this right had been granted by the Holy See and enjoyed for over 200 years (before 1370). This Constitution did not apply to the Oriental Catholic churches (e.g., Greek Catholics) which do not follow the Latin rite, nor to the rite of Zaïre created after Vatican II. Hence we may that before Vatican II the Latin Mass was the same all over the Catholic world, that is, in all Catholic churches of the Latin rite created between 1370 and 1965.
2.   See the web page of the CFRT, https://www.cfrt.tv/qui-sommes-nous/ (accessed on 9 August 2022).
3.   All interviews are available at https://www.lejourduseigneur.com/series/parole-inattendue-79 (accessed on 9 August 2022).
4.   All interviews are available at https://vodeus.tv/series/les-piliers-de-notre-dame-104 (accessed on 9 August 2022).
5.   Information given by an expert in church renovation during the discussion at *Le jour du Seigneur*, on 8 September 2019, but is not available online anymore.
6.   Part of this video can be seen at https://vodeus.tv/video/lart-de-la-paque-extrait-297 (accessed on 9 August 2022).
7.   The mass but not the interviews are available at https://www.youtube.com/watch?v=bH2tH_6zZnA (accessed on 9 August 2022).
8.   https://vodeus.tv/video/un-chemin-vers-la-lumiere-2379 (accessed on 9 August 2022).
9.   Personal recording. No public records.
10.   See note 9 above.

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
