# Peer review of "The Multicultural Church of “Le Jour du Seigneur”"

_religions, doi:10.3390/rel13080737_

Round 1

Author Response

I have read your report and appreciate your support.

Reviewer 2 Report

While I suspect there might be a thesis in this article - something to do with the deregulation of liturgical practices, multiculturalism, the importance of popular devotions, and the success of religious services as measured by the number of people in attendance - I still have a hard time working out what that thesis is. I still find the article to be excessively descriptive and not theologically articulate enough to constitute a scholarly piece of writing worth publishing. The author's quotation of Chinnici on p.3, lines 139-40, on the disconnect between the current form of the Roman rite and people's devotional needs suggests a rich area of study, but the article does not develop this connection enough. Likewise, the author introduces a question at the beginning of the conclusion on p.9, line 422 that might have served as a guide for a thesis statement that could be argued throughout the piece. But it comes only in the end, and it leads to yet more description. In the end, I really must say that I still do not know what the author is saying about the importance of multiculturalism, other than it seems to be very important. I am sorry to say that I still cannot see what this author means to say about the several lengthy descriptions it provides. The article is also awkwardly written and only scantly supported by its sources.

Author Response

“ I suspect there might be a thesis in this article.” No, there isn’t any. Sociologists mainly report factual findings, and only occasionally develop a sociological thesis.
My main point was described at the beginning of the third paragraph: “One reason liturgists should pay attention to this program is its outstanding success.” This paper is addressed to liturgists (who else can be interested in multicultural vs monoculgtural worship?)  This program is exceptional: I know of no other that is as successful as richly multicultural.  
Sorry it did not fit your expectations.

Reviewer 3 Report

The abstract does not introduce the research issue. It is only a description of a television program. No scratching of the research problem.

The article conceals an ecclesiological error concerning the understanding of the essence of the Church and the liturgy. According to the Catholic Church, the liturgy is not a private activity, but the worship of the Church, which is the sacrament of unity. Hence, liturgical activities belong to the whole Church. The liturgical rite has a strictly defined course, which is protected by liturgical law and ecclesiastical law so that no abuses are made and its essence is not changed. This requires humility and restraint on the part of all members of the Church. In fact, the most important thing in the liturgy is God and the worship of Him. Benedict XVI writes very carefully about it - "The Spirit of liturgy" or many other publications. It is very important to educate the faithful on the essence of the liturgy so that they participate consciously and that it is fruitful for them. Liberalization and individualization of the liturgy is an expression of a misunderstanding of its essence. The liturgy takes place in mortality, but is at the same time the pledge of the heavenly liturgy. Therefore, it cannot be treated as only a formal performance that should meet the expectations of the participants. This is clearly indicated in the Constitution on the Liturgy of the Second Vatican Council. Pius XII gives an important explanation of the liturgy in the special encyclical "Mediator Dei".

Therefore, the article lacks a clearly defined research goal. Is it about describing the liturgical practices presented in the program, which in the light of this article are not intercultural, but constitute a violation of the norms and the essence of the liturgy? This is perhaps the result of a lack of education and a strong tendency towards individualization and privatization of the liturgy, consuming it as a spectacle which, in fact, becomes only a human work. Or is the reception of liturgical participants examined in order to educate them properly through special TV programs?

In my opinion, the article contains a serious methodological and terminological error.

First, one should start with the definition of the liturgy in the Catholic Church, indicate documents and outline the post-conciliar discussions. Then define the concept of  multiculturalism and then examine selected programs in the light of the above definitions. Otherwise, predetermined conclusions are reached, which is not a proper research process.

Author Response

 “The article lacks a clearly defined research goal. Is it about describing the liturgical practices presented in the program, which... constitute a violation of the norms and the essence of the liturgy?”
Its goal is to describe liturgical practices. This paper is sociological and descriptive, nothing else. It may well constitute a violation of the essence of the liturgy, but such a criticism should be sent to the French bishops and the CFRT who consider the program to be orthodox according to their criteria. Who else is to judge? Not sociologists.
Sorry to have disappointed you.

Reviewer 4 Report

Minor corrections:
• l. 20: “one culture” (there is a letter missing)
• l. 67: “leave” instead of “live” (wrong word)
• l. 69: missing “the” between “was” and “position” (missing word)
• l. 76: “leaves” instead of “lives” (wrong word)
• l. 92: “credendi” instead of “credenda” (typo)
• footnote 1 shows an error
• l. 161: missing “of” between “half” and “the population” (missing word)
• l. 183: Consider giving the full name “Comité français de radio-télévision” already when mentioning the CFRT for the first time!
• l. 228: “is” instead of “being” (wrong word)
• l. 230: the “Pillars of Notre Dame” — the format you have used so far is to give the French title in italics and an English translation in parantheses (formatting)
• l. 240: the same
• l. 241: Byzantium is a proper name and should thus begin with a capital letter (spelling)
• footnote 5 shows an error
• l. 246: Byzantine/byzantine again
• l. 313 the “A Road towards Light” again as title and translated title (formatting)
• l. 315: missing “people” between “30” and “Among” (missing word)
• l. 317: missing “he” between “where” and “was” (missing word)
• l. 325–326: the French/English-title again
• l. 337: Who is Didier? I thought her name was “Brigitte”?
• l. 340–341: When was the day of migrants and refugees? And is it a publicly known and respected holyday?
• l. 341: “and” instead of “et” (wrong word)
• l. 371: Byzantine/byzantine again
• l. 372: Byzantine/byzantine again
• l. 374–375: missing quotation marks around the psalm verse
• l. 383: Byzantine/byzantine again

Comments:
• l. 19–24: This looks to me to be a very patriotic and optimistic introduction that might alienate especially non-US-readers. And… it doesn't apply here. Monoculturalism was also practiced in the 18th century in the new nation and is still today, see for example the (and I hate this term) “black churches”. I would suggest improving that section by using much more careful terms and by giving up the illusion of practiced multiculturalism anywhere in the world—OR: by adhering to the idea and supporting it with some references to widely accepted works and studies.
• l. 24–26: I understand what you want to claim here, but (1) You will have to define what you mean by writing "Liturgical churches" and (2) you miss the cases of inculturation of liturgy, e.g. the Zairian missal (which is of course monoculturalistic when seen as the missal of the Roman Catholic Christians within the church of the Democratic Republic of Congo, but is multicultural when seen as one of several... erm... "expressions" or "faces" of the one Roman Rite).
• l. 28–29: Please give some references for this list/typology! Did you develop it on your own?
• l. 31–33: Since it is apparently watched as live TV-show, the argument of global distribution would only apply to the nations living in a comparable time zone as Paris. I am uncertain if someone in the “New World” would watch this if it is broadcasted in the middle of the night.
• l. 38–39: The claim of organizational independency needs further references or further explanations: What is this group? Are they really independent or is there maybe a hint to some conflict of interest? Where does the funding come from?
• l. 42: Where does the number of viewers relate to? TV only? Or TV and online? What does the number say? Could you give something for comparison, e.g. the number of viewers of the regular evening news?
• l. 43: The claim that France is a “mainly agnostic country” needs at least a reference. The numbers I have show for the French Republic that a little more than the half is Catholic. And the national statistic commission mentions 58% of the French people believing in God.
• l. 45: The number of 45,000 attenders of Joel Austin’s Lakewood Church is not relatable. You compare the watchers of TV and online on the one hand to the direct and personal attenders at a certain place on the other hand.
• l. 53: “lex orandi lex credendi (the law of prayer is the law of faith)” is not what Prosper said (or wrote). His argument was "ut legem credendi lex statuat supplicandi" (so that the law of prayer judges over the law of faith) (DH 246) and it was part of his eighth argument in an anti-semi-Pelagian writing. His argument which he unfolds there is based on wrong assumptions but even more important is that he never argued that this is an equation that would work in both directions as well as that this could somehow be a universal principle.
• l. 55–56: The second idea that the doctrines inform the prayer is thus not possible. It is only possible if you quote Prosper wrong.
• l. 56–57: How do you come to your conclusion “The first favors multicultural worship and the second monocultural uniformity.”? I was not able to follow your argument here.
• l. 59–61: Regarding your statement that the Council of Trent defined the liturgy of the Roman Missal exclusively as liturgy, it appears to be necessary to give references for this. I cannot find this idea within the resolutions of the Tridentince Council. It seems to me that the split between "private worship" and "official liturgy" began much more earlier (Amalarius?) and ended much more later (Second Vatican Council?). There is a lot you can blame the Tridentinum for, but not for narrowing the meaning of the term "liturgy".
• l. 67: You wrote that you will leave the historical setting aside and you refer to Novak. I guess the historical context would help understanding why we misunderstand Prosper today. It is worth giving a few words about this.
• l. 69–71: Here you quote Pope Pius XII but you missed giving the reference for this quotation. Please add.
• l. 73: The “lex credendi legem statuat supplicandi” is nothing that Pius XII invented, but what Prosper actually wrote!
• l. 76–77: Regarding the position of Pope Pius XII I must note that there were other popes before and after him. There was a lot of missionary work and there were a lot of documents on inculturation. It seems odd to me to quote one pope here, which in fact was a progressive thinker in favor of the Liturgical Movement, to prove a universal principle of the Roman Catholic Church. Furthermore: Please note that the Oriental Catholic Churches were allowed to hold on to their own rites (since Benedict XIV.) So, at the time of Pius XII writing the "lex supplicandi" of the Catholic Church was already diverse.
• l. 82: Regarding Kavanagh’s concept of theologia prima please note the discussion about the concept itself and the manifold prejudices which are hidden therein. For further information read Bradshaw, Paul F.: Difficulties in Doing Liturgical Theology. In: Pacifica 11 (1998), 181–194, esp. 192f.
• l. 94: You stated that Irwin’s concept had little impact on the worshipping practice. I don't think that it is the aim of liturgical theologians to write books that have an effect on the non-academic public. You are speaking here of concepts to describe reality - as soon as it affects reality it is no description.
• l. 97–98: You mention that Vatican II introduced the division of liturgy and pious exercises. Please add the reference to SC 13 and consider discussing the section here.
• l. 102–104: Your phrase “With the publication of the first Roman Missal in 1570, only the liturgical practices mentioned in the Missal would be officially recognized by the church.” is wrong. In the sixth paragraph of his bull "Quo Primum" (the paragraph begins "ut autem a sacrosancta") Pius V. orders that the Tridentine Missale replaces all the missals younger than 200 years. All the others are still recognized. Besides this the congregatio pro propaganda fidei was always allowed to recognize liturgical rites for the mission that have no roots in the Roman liturgical books.
• l. 104–105: Regarding the official book of prayers I was wondering what you would suggest to be the official book of prayers of Lutherans or the Reformed Church in the 16th century? Almost every local parish had its own ritual book and for a long time there was no such a thing as an official book of prayers in the Protestant Churches.
• l. 134–136: Please give a reference for the US critic speaking about “piety void”!
• l. 143: What is “Catholic decline”? Do you mean the decline of the number of practitioners within the US-society?
• l. 143: The sentence “Yet there is no return to the past.” is pure rhetoric and can be deleted.
• l. 147–148: You assumed that parishioners derivate from the liturgical norms “without theological, canonical or conciliar arguments”. I know that Martin Stuflesser had a project on this topic (deviations from the normed pratice) in Germany. And it turned out that most of the ministers reflected what they are doing and were able to give theological arguments. Is there something similar for the USA or France? Anyway, the instruction "Redemptionis sacramentum" pushes priests to obeying the liturgical books by encouraging lay people to call the higher instances in case of abuses of liturgy (it is remarkable how often John Paul II uses the term "abusus" in this document when considering how often he avoided it in other cases).
• l. 149: Pastors don’t “enjoy a fair amount of freedom”, especially not since the promulgation of “Redemptionis Sacramentum”.
• l. 154: Regarding your assumption that the whole service “would be considered liturgical” you missed telling by whom. I guess, there are differences.
• l. 168–169: When writing that the French state covers the half of the production costs you should also write how the other half is covered. I assume that the CFRT is not that independent as it should be.
• l. 184: “It is the ultimate authority of the organization”—which organization?
• l. 184–185: I like the sentence “It can only function productively if there is collaboration between lay and clerical views.”—but I doubt it. The committee has nine members which is an odd number, which means that either the clerics or the lay people are the majority.
• l. 188–189: How can the committee be lay if at least three monks are part of it?
• l. 196: the number of 9 members but 60 employees raises doubts if I understood properly what is designated by the term “CFRT”. Please specify!
• l. 208: When referring to the title of Parole inattendue you miss to inform about the object: Is it the title of the TV show or the title of a section within the TV show?
• l. 212–216: The list doesn’t need to be that long to demonstrate what you mean.
• l. 221–223: Instead of writing about “most” and “few” please indicate the exact numbers and where they can be verified.
• l. 223–224: When reporting about people expressing face or belief in Christ etc. you should give a reference to your source!
• l. 229: The question of non-believers invites to further investigation and examination of Karl Rahner’s concept of “anonymous Christianity”. ;-)
• l. 232: I was wondering about the fire at Notre Dame of 1915. Do you mean the fire of 2019?
• l. 235–237: If the purpose of the series is your assumption, please indicate so! Or did you get this from some kind of "mission statement"? Then, please, add a reference, too!
• l. 241: Since I do not know the TV program it would be better first to tell what it is about before telling who was taking part
• l. 246–260: What is the purpose of the long section? It doesn't help to get to your results.
• l. 262–263: Is the variety of addressed public your impression or do they somewhere indicate their mission statement?
• l. 269: What do you understand to be the "content of LJS" if not the magazine nor the mass?
• l. 273–276: This is a very low material basis. Did you try to contact the TV-channel and to ask for copies for research purposes?
• l. 303: On which date exactly was the anniversary mass?
• l. 362: I guess the “for you but without you” is nowhere written in the "Tridentine Missal". Rather it sounds polemic. Again, you should at least refer to articles or books treating the question of personal attendance at the post-Medieval mass. It’s not that simple as you present it here.
• l. 371–373: Neither the icon of Christ nor the Trisagion are purely Byzantine, but widespread among all the Oriental Churches. Please specify what you mean by writing "Byzantine"!
• l. 389: How do you know why they chose those churches if it was not displayed on the screen? And do you also know the reasons for this decision?
• l. 390–392: There is no need of that list. If you want, you can put it in the footnotes.
• l. 408–409: Why do you assume that it would be shocking if a lay person would grab a microphone “in any Catholic parish”? Did you try to do so?
• l. 424–436: Please consider giving the historical background of the program much earlier in this article when introducing it.

Author Response

I am very grateful for your extensive review with 25 corrections and about 50 queries or criticisms. Your keen eye was able to see typos which I did not see after multiple proof reading.
I am less enthusiastic about your criticisms which are mainly negative, starting in the first sentence. “This [is an] introduction that might alienate especially non-US-readers. And… it doesn't apply here.” Why would it alienate non-US-readers  and probably yourself? And where is “here?”  I cannot see why a worship program involving Protestants and Catholics, lay people and clerics, academicians and religionists, and liturgies taking place all over the world would not be applicable  to “here?”
You point out that this TV show applies only to the time zone of Paris. If so, does it make it less valuable?  “I am uncertain if someone in the ‘New World’ would watch this if it is broadcasted in the middle of the night.”  Because it is available on the internet, I had no problem watching it at any time.
You include theological disputes, like, “lex orandi lex credendi... is not what Prosper said (or wrote). His argument which he unfolds there is based on wrong assumptions but even more important is that he never argued that this is an equation that would work in both directions as well as that this could somehow be a universal principle.” I am aware of the various interpretations of the lex orandi, but a sociological description of a TV program is not the place to introduce historical and theological issues. This paper is geared to liturgists, not historians or theologians.
I am sorry to have disappointed you, but I appreciate your thorough review.

Round 2

Reviewer 4 Report

Just a few minor suggestions/remarks:

• l.24–25: The Latin Mass was, in fact, NOT the same all over the world. On the one hand, the churches of Milano and Toledo were allowed to keep their proper mass rite and on the other hand all of the Oriental Catholic churches, among them the huge Greek-Catholic Church, kept celebrating in a non-Latin rite. In some cases, nations were allowed to adapt the mass to their culture, the most prominent of them the mass rite of Zaïre (Congo), which even today includes dancing and the invocation of the ancestors (you have this in your contribution in l. 500–512). I would suggest to add an “almost” or a “for the most part” in this sentence.

• l. 25–26: Please give examples for the strong resistance toward multicultural worship. I guess there are a lot and it would suffice to give maybe two or three within the footnotes. You will surely find something to quote in “Redemptionis Sacramentum” or “Liturgiam Authenticam”.

• l. 30–33: These sentences sounds flippant to me. The reasons behind the different struggles can’t be reduced to a conflict between Shiite and Sunnite but also include economic and ethnic issues. Furthermore, there are also Christians taking part in the conflicts. Please make sure that the Shiite/Sunnite thing or the Rohingya thing are in each case one of the explanations that media and/or fanatics offer to us, and which give us a feeling of being a purely religious conflict.

• l. 46–54: This paragraph helps a lot! I like it.

• l. 67: I guess you mean Central European Time, but it would help to add this information.

• l. 72–73: I guess this is one issue were we maybe misunderstood each other or where we do not see the other’s issues. Basically, my question is: Is the worship from “a different parish anywhere in the world” a live-stream or a pre-recorded worship? And the answer to this affects the question if people all over the world celebrate together at the same time in one huge worship or if they rather watch the same TV-show at different times in different places. At least from a theological point of view there is a huge difference between a live-streaming and a pre-recorded celebration.

• l. 104–105: It is not only the dichotomy between “official liturgy” and “devotion”. Within “official liturgy” there were uniform parts as for example the texts or the clerical ranks but also culturally adapted parts as for example the architecture of the building and the paintings. Most interesting in this case is the music which started as the uniform Gregorian chant but soon developed to different kinds of church singing with huge differences e.g. between France, Italy and Germany.

• l. 223: Consider highlighting the URL

• l. 232: “Parole inattendue” is the title to what? You haven’t mentioned it so far.

• l. 403: Add “of” between “singing” and “a”

Author Response

“The singularity of the pre-Vatican II Latin Mass.” Reply: the Constitution Quo Primum of Pius V (1570) recognizes the right “to celebrate Mass differently” when this right had been granted by the Holy See and enjoyed for over 200 years (before 1370).  The Tridentine Liturgy does not apply to the Oriental Catholic churches (e.g., Greek Catholics) which do not follow the Latin rite, nor to the rite of Zaïre created after Vatican II. Therefore, my writing that "Before Vatican II, the Latin Mass was the same all over the Catholic world" refers to all Catholic churches of the Latin rite created between 1370 and 1965, that is, 99.99% of the parishes in the world. But to make the text less absolute, I will change “the same” to "almost the same."

“Please give examples for the strong resistance toward multicultural worship.” Reply:  replace, “Before Vatican II, the Latin Mass was the same all over the Catholic world. As a consequence, there is often today a strong resistance toward multicultural worship” with “Before Vatican II, the Latin Mass was almost the same all over the Catholic world. The Vatican II liturgical reforms to overcome this monoculturalism were faced with strong resistance which continue to this day.  Archbishop Marcel Lefebvre opposed most Vatican II innovations. He created the Society of Saint Pius X from which emerged The Society of Saint Pius V, the Priestly Fraternity of Saint Peter, and numerous other resistance groups.

“The conflict between Shiite and Sunnite also includes economic and ethnic issues.” Reply: to the sentence “In the Middle East where Christians and Muslims have cohabited peacefully for centuries, the Shiites and the Sunnis kill one another.” I have added, “Of course, these conflicts also involve economic and ethnic tensions.”

“My question is: Is the worship from “a different parish anywhere in the world” a live-stream or a pre-recorded worship?” Reply: to the sentence, “It offers the Sunday mass for believers, but each week in a different parish which stimulates interest.” Add: “This mass is broadcast live Sunday morning for television viewers, and in recorded form a few hours later for internet viewers.”